# The Anti-Acne Potential and Chemical Composition of *Knautia drymeia* Heuff. and *Knautia macedonica* Griseb Extracts

**DOI:** 10.3390/ijms24119188

**Published:** 2023-05-24

**Authors:** Małgorzata Chrząszcz, Małgorzata Miazga-Karska, Katarzyna Klimek, Michał P. Dybowski, Rafał Typek, Dorota Tchórzewska, Katarzyna Dos Santos Szewczyk

**Affiliations:** 1Department of Pharmaceutical Botany, Medical University of Lublin, 1 Chodźki Str., 20-093 Lublin, Poland; malgorzata.chrzaszcz329@gmail.com; 2Department of Biochemistry and Biotechnology, Medical University of Lublin, 1 Chodźki Str., 20-093 Lublin, Poland; malgorzata.miazga-karska@umlub.pl (M.M.-K.); katarzyna.klimek@umlub.pl (K.K.); 3Department of Chromatography, Institute of Chemical Sciences, Faculty of Chemistry, Maria Curie Sklodowska University in Lublin, 20-031 Lublin, Poland; michal.dybowski@poczta.umcs.lublin.pl (M.P.D.); rafal.typek@poczta.umcs.lublin.pl (R.T.); 4Department of Plant Anatomy and Cytology, Maria Curie-Skłodowska University, Akademicka 19, 20-033 Lublin, Poland; dorota.tchorzewska@poczta.umcs.lublin.pl

**Keywords:** *Knautia*, Caprifoliaceae, skin disorders, antibacterial, anti-inflammatory, antioxidant, cytotoxicity, LC-MS, GC-MS

## Abstract

The treatment of acne and other seborrheic diseases has arisen as a significant clinical challenge due to the increasing appearance of multi-drug resistant pathogens and a high frequency of recurrent lesions. Taking into consideration the fact that some *Knautia* species are valuable curatives in skin diseases in traditional medicine, we assumed that the thus far unstudied species *K. drymeia* and *K. macedonica* may be a source of active substances used in skin diseases. The purpose of this study was to evaluate the antioxidant, anti-inflammatory, antibacterial, and cytotoxic activities of their extracts and fractions. An LC-MS analysis revealed the presence of 47 compounds belonging to flavonoids and phenolic acids in both species while the GC-MS procedure allowed for the identification mainly sugar derivatives, phytosterols, and fatty acids and their esters. The ethanol as well as methanol-acetone-water (3:1:1) extracts of *K. drymeia* (KDE and KDM) exhibited great ability to scavenge free radicals and good capacity to inhibit cyclooxygenase-1, cyclooxygenase-2, and lipoxygenase. Moreover, they had the most favorable low minimal inhibitory concentration values against acne strains, and importantly, they were non-toxic toward normal skin fibroblasts. In conclusion, *K. drymeia* extracts seem to be promising and safe agents for further biomedical applications.

## 1. Introduction

Among all skin diseases, which are marked by an abnormal function (usually hyperactivity) of the sebaceous glands in the skin, acne vulgaris is the most common, with prevalence in 99% of the acne cases. Every seborrheic disorder, particularly acne vulgaris, is a chronic disease with a multifactorial etiology and a bacterial and inflammatory basis [1].

Acne vulgaris is one of the most common skin disorder among all dermatological diseases involving sebaceous glands [2]. Adolescents and young adults are most affected, and the incidence of acne is higher in young men [3,4,5]. It is estimated that approximately 85% of the human population has suffered from acne at least once through their lifetime [5,6].

Four important processes are involved in the pathogenesis of acne: the overproduction of sebum with an altered composition due to increased androgen levels, hyperkeratisation of pilosebaceous duct, resulting in colonization by *Cutibacterium acnes* (=*Propionibacterium acnes*), which in turn leads to inflammation [7,8].

Due to androgen overproduction and increased sebum production, *P. acnes* release various hydrolytic enzymes, causing the breakdown of sebum into free fatty acids and glycerol. The free fatty acids act similarly to chemokines, resulting in the release of pro-inflammatory factors, such as interleukin 8 (IL-8) and tumor necrosis factors (TNF-α). In addition, the accumulation of cytokines provokes macrophages to migrate into the inflamed area. Another consequence of hydrolytic enzymes is oxidative damage to the hair follicle wall and the release of free radicals [9,10].

Due to the multidirectional etiology of seborrheic disorders, effective treatment should also be multidirectional. The therapeutic substance inhibits sebum secretion, reduces follicular hyperkeratisation, and has an antibiotic-like effect, but also prevents skin disfunction, such as acne vulgaris [2].

With the increasing phenomenon of multidrug resistance (MDR) resulting in ineffective therapy and recurrent skin lesions, it is crucial to find an effective cure [11,12].

The problem of MDR results from the overuse of broad-spectrum antibiotics, such as tetracyclines, to treat every type of acne lesion, leading to increased resistance among bacterial pathogens [11].

It is therefore worth emphasizing that the widespread use of antibiotics is harmful and needs a new approach to the treatment and prevention of skin diseases based on safer plant-based agents [2].

Due to the presence of traditional medicine, we have access to reliable knowledge and unlimited resources of botanical material. Therefore, scientists have started to revise ethnopharmacology to develop new, more effective, and side-effect-free drugs for the treatment of a wide variety of skin diseases related to seborrheic disorders [13].

However, many aspects of the action of plants used in the treatment of seborrheic diseases are still unresolved and many questions about their roles in these disorders remain open. Therefore, further research in this important area is warranted.

In our study, we chose to investigate two *Knautia* species belonging to the Caprifoliaceae family in this direction.

The idea of our research results from a deep study of the existing literature and the lack of much information on the species studied. We focused on proving the antioxidant, antibacterial, and anti-inflammatory effects while demonstrating the safety of *Knautia* extracts for human skin fibroblasts. These effects could potentially find application in the treatment of various types of acne vulgaris or other disorders resulting from the overproduction of sebaceous glands. The genus *Knautia* (Caprifoliaceae) comprises 48 species, which are mainly found in parts of Europe, Asia, and Africa [14].

In the literature, there are only few reports regarding the chemical composition and biological activity of these species. However, it has been reported that extracts of many different *Knautia* species contain, among others, phenolic acids, the presence of which may be responsible for their antioxidant and antimicrobial activity [15,16]. 

In addition, an infusion of *K. arvensis* herb was found to have anti-inflammatory, expectorant, diuretic, and analgesic properties [17].

Moreover, in the past, many plants belonging to this family, such as *Lonicera japonica* Thunb. or *Scabiosa columbaria* L., were used to treat seborrheic skin disorders [2,18,19,20]. In addition, in our previous study, we have shown that *Cephalaria* species have the ability to inhibit the growth of many bacterial strains that cause skin diseases and inflammatory lesions, as well as scavenge reactive oxygen species (ROS) [2]. These facts are very important and needed in a new approach to the treatment of multidirectional skin disorders.

Plants belonging to the Caprifoliaceae family are rich in a variety of chemical compounds, such as phenolic acids [21], flavonoids [22], and saponins [23], which may be responsible for their activity. 

## 2. Results and Discussion

### 2.1. Chemical Composition of the Extracts

#### 2.1.1. Liquid Chromatography–Mass Spectrometry Analysis (LC-MS)

In the first step of our study, the chemical composition of the extracts obtained from *Knautia drymeia* and *K. macedonica* was investigated using the LC-MS method. Appendix A shows 47 identified compounds, including their molecular formula, theoretical and experimental molecular mass, both errors in ppm and mDa, and the fragments.

In our study, mainly flavonoids and phenolic acids were identified using an LC-MS analysis. Exemplary chromatograms with marked main peaks are displayed in Appendix A. The results of the quantitative analysis are shown in Table 1 and Table 2. The quantities of compounds identified were carried out on the basis of the calibration curves obtained for the standards. For the quantitative analysis of compounds that do not have standards, calibration curves for substances of a similar structure were used.

The phytochemical analysis using LC-MS method of extracts and fractions of *K. drymeia* and *K. macedonica* provided information on 47 detected active compounds. The compounds were characterized and described on the basis of their UV-VIS spectra and MS/MS spectra. It was found that the acetate fractions of methanol–acetone–water extracts (KMM-O—143,175.8 μg/g DE and KDM-O—128,484.7 μg/g DE) contain the largest amount of phenolic compounds. The smallest amounts were found in the butanol fraction of methanol–acetone–water extract of *K. macedonica* (KMM-B—86,797.4 μg/g DE) and ethanol extract of this species (KME—88,569.2 μg/g DE). It is worth noting that in all extracts and fractions, phenolic acids constituted from 71.77 (KMM-O) to 83.31% (KDM-B) of the total content of compounds. The discovered compounds belong to two groups: phenolic acid derivatives (16 compounds detected) and flavonoids (31 detected). The first, smaller group were caffeic acid derivatives, which include, inter alia, chlorogenic acid, neochlorogenic acid, cryptochlorogenic acid, caffeic acid, and 3,5-di-*O*-caffeoyllquinic acid. 3,5-Di-*O*-caffeoyllquinic acid (2188.5—31,560.2 μg/g DE) was present in the highest amount, especially in the butanol fraction of the KDM extract (31,560.2 μg/g DE) and the ethyl acetate fraction of the methanol–acetone–water extract of *K. macedonica* (KMM-O; 18,885.1 μg/g DE). In contrast, its lowest amount was recorded in the butanol fraction of the *Knautia macedonica* extract (2188.5 μg/g DE). The caffeoylquinic isomer has also been previously described in other plants belonging to the Caprifolicaeae family [2,21]. Chrząszcz et al. described the presence of two caffeoylquinic derivatives in extracts from *Cephalaria uralensis* and *C. gigantea* [2].

The second compound detected in the highest amount was chlorogenic acid, with the highest levels in the butanol (28,043.0 μg/g DE) and the ethyl acetate (27,015.6 0 μg/g DE) fractions of *Knautia macedonica* extract. Ethanol extracts of both species were the poorest sources of this compound (21,256.1 μg/g DE-KME and 21,845.2 μg/g DE-KDE). Extracts of *K. arvensis* herb prepared in Poland and France also showed the presence of chlorogenic acid [21,24]. Two more identified acids belonging to the chlorogenic acid derivatives are cryptochlorogenic acid and neochlorogenic acid. The extracts and fractions richest in these constituents were the ethanol extracts from *K. drymeia* and the ethyl acetate fraction from *K. macedonica*, respectively. Various extracts from *C. uralensis* and *C. gigantea* are also good sources of cryptochlorogenic acid, in which they have been proven to be present in large quantities [2]. All the extracts and fractions of both species tested had the lowest content of neochlorogenic acid, which is in agreement with a previous study on an ethanol extract from *K. arvensis* [25].

The next group of identified phenolic acids were benzoic acid derivatives and included *p*-hydroxybenzoic acid (311.2–6165.2 μg/g DE), *m*-hydroxybenzoic acid (141.2–524.3 μg/g DE), gallic acid (16.4–476.2 μg/g DE), vanillic acid (548.0–1812.4 μg/g DE), and salicylic acid (370.4–3112.2 μg/g DE). A similar content was also described by Karailja et al. in an ethanol infusion of aerial parts of *K. arvensis* [22].

Hydroxycinnamic acid derivatives were also detected in our study, including ferulic acid (1988.7–5960.4 μg/g DE), sinapic acid (0.5–1284.0 μg/g DE), and *p*- (1.9–94.4 μg/g DE) and *o*-coumaric acid (2.2–100.4 μg/g DE). Ferulic acid was detected earlier in shoots of *K. sarajevensis* (Beck) Szabó [26] while sinapic acid and *o*-coumaric and *p*-coumaric were detected in the aerial parts of *K. arvensis* [22,25].

It has been reported that phenolic acids exhibit strong antioxidant and antimicrobial activity [2], which can be beneficial for the *Knautia* species extracts and fractions evaluated in this study.

The flavonoids in this study represent a much larger group of compounds occurring in total, as 31 compounds from this group were detected in the extracts and the fractions obtained from them. Kaempferol and its four derivatives (kaempferol-3-*O*-rutinoside, kaempferol-3-*O*-glucuronide, kaempferol-7-*O*-glucoside, and kaempferol-3-*O*-glucoside) were detected in each extract and fraction. Kaempferol derivatives were previously described in the aerial parts of *K. integrifolia* by Giambanelli and co-authors [27].

Among luteolin and its derivatives, luteolin-7-*O*-rutinoside (18.8–712.3 μg/g DE), luteolin-7-*O*-glucuronide (133.2–880.2 μg/g DE) and luteolin-7-*O*-glucoside (153.2–4360.2 μg/g DE) were identified. These compounds had previously been found in the aerial parts of *K. arvensis* and the leaves of *Lonicera caerula* [27,28].

The next group of flavonoids found in the extracts and fractions were quercetin derivatives. The compound identified in greatest quantity was quercetrin (804.2–1896.6 μg/g DE), which had not previously been detected among *Knautia* species. The second compound in large amounts, quercetin, is a common secondary metabolite in the plant world. Among *Knautia* species, it has previously been described in an extract from the shoot [15] and the aerial parts of *K. sarajevensis* [26]. Isoquercitrin (quercetin 5-*O*-glucoside; 49.6–215.2 μg/g DE) was also detected in all extracts and fractions examined. Previous studies have reported the presence of this flavonoid in the fruit extract of *Lonicera nigra* and *L. xylosteum* [29]. Quercetin derivatives, such as quercetin-3-*O-β*-D-(2″-*O-β*-D-xylosyl)galactoside and quercetin-3-*O*-glucuronide, were also found in extracts and fractions from *K. drymeia* and *K. macedonica*, which were previously described in extracts of the aerial parts of *K. integrifolia* [27].

Among the flavone derivatives, we detected three compounds in the extracts and fractions studied: apigenin (28.7–341.6 μg/g DE), naringenin (24.4–1053.8 μg/g DE), and chrysin (20.6–198.4 μg/g DE). Apigenin was also detected in *K. montana* flowers in an earlier study [30]. Naringenin and chrysin, on the other hand, were detected in the methanolic extract of the aerial parts of *K. sarajevensis* [26].

Galangin (flavonol; 7.5–86.8 μg/g DE) and pinocembrin (flavanone; 60.8–912.5 μg/g DE), present in the extracts tested, were previously found in the shoots and aerial parts of *K. sarajevensis* [15,26].

It is worth underlying that flavonoids are very important factors that protect the cell against oxidation, so their occurrence can affect the strong antioxidant activity [31].

#### 2.1.2. Gas Chromatography–Mass Spectrometry Analysis (GC-MS)

The second method used to identify active compounds in extracts and fractions from *K. drymeia* and *K. macedonica* was the appropriate GC-MS method (Table 3 and Table 4, and Appendix A, respectively). Among the compounds detected, the leading groups were sugar derivatives, phytosterols, fatty acids and their esters, and volatile compounds. The dominant group of compounds in the extracts and fractions from *K. drymeia* were sugar derivatives. The content of methyl β-D-glucopyranoside ranged from 31.18% to 56.72%. In the butanol fraction of the methanol–acetone–water extract of *K. drymeia*, the content of methyl α-D-glucopiranoside exceeded 79%. In the case of extracts and fractions from *K. macedonica*, the amount of methyl β-D-glucopiranoside is noteworthy as well. The content of this compound ranged from 28.26–64.31%. Only the ethyl acetate fraction did not show the presence of this compound.

The second dominant group of compounds contained in the extracts and fractions were fatty acids and their esters. The most important ester identified included octadecanoic acid, 2,3-dihydroxypropyl ester, which was 10.16% in the ethanol extract of *K. macedonica* (KME), 9.85% in the methanol–acetone–water extract of *K. macedonica* (KMM), 7.13% in the butanol fraction of the KMM, and 7.11% in the KDM.

The third major group of compounds in both species were phytosterols. Among the most important was campesterol (which accounts for 12.98%—KME; 9.16%—KDE; 8.77%—ethyl acetate fraction of KMM). Other noteworthy compounds were 2-dodecenal (11.82% in KDE), *cis*-7-tetradecen-1-ol (8.06% in KDE), and *d*-gluco-heptulosan (11.49% in KMM).

Previous studies have also detected the presence of phytol and palmitic acid in extracts from *K. arvensis* [26], but their amounts in the tested extracts and fractions from *K. drymeia* and *K. macedonica* did not exceed 2%.

It is worth mentioning that campesterol found in each of the extracts and fractions tested, has been previously described as an anti-angiogenic compound [32].

The wide range of active compounds identified in the extracts of *Knautia drymeia* and *K. macedonica*, leads to an overlapping effect of different agents, making it possible to act on several levels of different oxidative stress-induced diseases.

### 2.2. Biological Activity

#### 2.2.1. Antioxidant Activity

Antioxidants are a group of chemical compounds that remove the excess of free radicals and thus prevent the development of many civilization diseases, as they support the natural defense mechanisms of human cells. Plants are a rich source of antioxidants due to the content of i.a. polyphenols.

The antioxidant activity was studied on a microplate scale in cell-free systems. The *Knautia* extracts and fractions were evaluated at a concentration ranging from 0.078 to 10 mg/mL. It was shown that all tested extracts and fractions exhibited moderate scavenging capacity in a concentration-dependent manner (Table 5). For comparison, the radical scavenging activity of ascorbic acid (AA; IC_50_ = 0.48 ± 0.30 mg/mL) was tested in the same conditions. The highest DPPH scavenging activity was shown for the KDM extract (IC_50_ = 0.50 mg/mL), and this activity was similar to that of ascorbic acid. High activity was also found for the KDE and KDM-B extracts (IC_50_ = 1.36 and 1.67 mg/mL, respectively) while the weakest activity was noted for the KME extract (IC_50_ = 4.04 mg/mL, respectively).

As shown in Table 5, similar to the DPPH test, the ABTS^●+^ assay revealed that the KDM extract possessed the strongest ability to scavenge free radicals (IC_50_ = 0.41 mg/mL), followed by the KDE (IC_50_ = 0.46 mg/mL) and the KDM-B (IC_50_ = 0.50 mg/mL) extracts. The lowest activity was recorded for the KMM and KME extracts.

As reported in Table 5, the extracts and fractions from both species tested had the ability to interfere with the formation of iron and ferrozine complexes, suggesting their chelating capacity and ability to capture iron ions before ferrozine. The KDM extract showed the highest activity (IC_50_ = 0.06 mg/mL), and this was similar to the positive control—Na_2_EDTA*2H_2_O (IC_50_ = 0.04 mg/ML).

Karalija et al. [15] used specific peroxidase activity and DPPH assay to evaluate the antioxidant activity of methanol shoot extracts of *K. sarajevensis* multiplied in Murashige and Skoog (MS) liquid medium containing 6-benzyladenine—BA, kinetin—KIN, and Zeatin—ZEA. DPPH assay showed that radical scavenging activity of shoot cultures ranged from 10 up to 90 µg/mL.

In our earlier study, we evaluated the antioxidant activity of *C. gigantea* and *C. uralensis* extracts [33]. The higher DPPH^•^ scavenging activity was found for the aerial parts of *C. uralensis* (IC_50_ = 2.86 ± 0.12 mg/mL), which was comparable to the activity obtained for KMM-O in this research. On the other hand, *C. uralensis* flower extract was found to be the strongest free radical scavenger (IC_50_ = 0.45 ± 0.21 mg/mL).

#### 2.2.2. Enzyme Inhibitory Activity

Due to the fact that *Knautia arvensis* herb in the form of infusions, broths, and alcoholic infusions has long been used in traditional medicine to prepare of anti-inflammatory agents for the treatment of skin diseases [25], in the following, we tested the anti-inflammatory activity of extracts and fractions of *K. drymeia* and *K. macedonica*.

To determine the potential anti-inflammatory properties of the *Knautia* species, we estimated the ability of the extracts and fractions to inhibit the conversion of arachidonic acid to PGH_2_ by ovine COX-1 and human recombinant COX-2 using a COX inhibitor screening kit (Cayman Chemical, Ann Arbor, MI, USA). As shown in Table 6, most of the studied extracts and fractions showed fairly good activity against COX-1 and COX-2. The most active against COX-1 were KDE and KDM extracts (IC_50_ = 7.63 and 14.99 µg/mL, respectively), followed by KDM-B and KMM-B fractions (IC_50_ = 29.19 and 30.18 µg/mL, respectively), while the weakest was the ethanol extract of *K. macedonica* (IC_50_ = 53.76 µg/mL). We obtained similar results against COX-2. The most active extracts were KDE and KDM (IC_50_ = 10.68 and 11.65 µg/mL), but their activity was almost three times weaker than that of indomethacin used as a positive control (IC_50_ = 3.90 µg/mL).

In our study, we also tested the lipoxygenase inhibition by the *Knautia* species because lipoxygenases (LOX) play a key role in stimulating inflammatory responses [34]. The results of the inhibition of lipoxygenase are shown in Table 6. Only the ethanol extract of *K. drymeia* (KDE) showed a considerable ability to inhibit lipoxygenase activity (IC_50_ = 14.46 μg/mL); however, this activity was still more than two times weaker than that of the positive control (IC_50_ = 6.05 μg/mL).

#### 2.2.3. Antibacterial Activity

The data shown in the Figure 1 clearly indicate that the most active against acne bacteria were extracts from *K. drymeia*, mainly the fractions KDM-O and KDM-B, for which the inhibition zones were 25–23 mm and 16 mm, respectively, and for the crude methanol–acetone–water extract of *K. drymeia* (KDM), of which the zones were 16–15 mm. These extracts were the strongest among all samples to inhibit the growth of aerobic Gram-positive bacteria, reaching zones in the range of 22–10 mm. Significant activity against both microaerobic acne strains and aerobic Gram-positive strains was shown by the ethyl acetate fraction of the methanol–acetone–water extract of *K. macedonica* (KMM-O), for which the growth inhibition zones of these bacteria were 23–22 mm and 18–13 mm, respectively. However, the butanol fraction of this extract (KMM-B) and also crude methanol–acetone–water extract of *K. macedonica* (KMM) showed only moderate activity against Gram-positive strains, including acne ones. Ethyl acetate fractions (i.e., KDM-O and KMM-O) moderately (8–10 mm) inhibited the growth of Gram-negative *E. coli* while the other samples did not have such activity. Stronger antibacterial activity was shown by the separated fractions in comparison with the raw extracts of the tested plants.

Table 7 confirms that the most favorable low minimal inhibitory concentration values of all tested samples against acne strains were obtained from the methanol–acetone–water extract and its fractions from *K. drymeia* (KDM, KDM-O, KDM-B), which were in the range of 750–1500 µg/mL, then for KMM-O and KMM-B at 1500–6000 µg/mL. The remaining samples were not tested due to low activity in the screening test or no recorded MIC values in the concentration range used.

The MBC/MIC ratio values > 4 obtained for each of the tested samples—regardless of the family and the type of extract fraction—prove that they had a bacteriostatic and not bactericidal character.

In previous studies on the antimicrobial activity of *Knautia* species, only *K. arvensis* and *K. sarajevensis* has been tested in this direction [15,35]. Karalija et al. [15] reported that methanol extracts of shoots of *K. sarajevensis* cultivated in media containing 2.0 mg/L 6-benzyladenine were moderately active against *Staphylococcus aureus* and *Bacillus spizizeni*.

Sarıkahya et al. [36] also showed that the newly isolated from endemic *Cephalaria* species in Anatolia (Turkey), triterpene glycosides scoposides F and G and paphlagonoside A, paphlagonoside B, and isacoside, exhibit as strong antibacterial activity against Gram (+) and Gram (−) bacteria.

Moreover, a study of the antimicrobial activity of essential oils from fresh fruits, herbs, and flowers of *Scabiosa arenaria* Forssk. was performed against Gram-(+) bacteria (e.g., *Streptococcus* spp., *Staphylococcus saphrophyticus*, *S. aureus*, *S. epidermidis*) and Gram-(−) bacteria: *Escherichia coli*, *Klebsiella pneumoniae*, *Pseudomonas aeruginosa*, *Citrobacter diversus*, *Salmonella enteritidis*, *Acinetobacter baumanii*, *Enterobacter cloacae*, *Serratia marcescens*, as well as pathogenic fungi of the *Candida* genus. The study showed that flower essential oil had the strongest effect, followed by herb oil, and fruit oil with the least. The activity of the essential oil from flowers against *S. aureus* (MIC = 0.1562 mg/mL; MBC = 0.3125 mg/mL) was slightly lower than thymol (MIC = 0.2 mg/mL) and higher than gentamicin (MBC = 0.01562 mg/mL) used as reference substances. In relation to other microorganisms, flower oil showed moderate activity, similarly to herb and fruit essential oil. The antimicrobial activity against *S. aureus* of the fruit oil was significantly weaker than that of the herb oil while against *E. cloacae* and *S. marcescens*, it was stronger [37].

#### 2.2.4. Cytotoxic Activity

After 72 h of incubation, the MTT assay showed that some of the tested extracts were non-cytotoxic towards BJ cells (Figure 2). Thus, KMM-B, KDE, and KDM-B extracts, even at the highest tested concentration (i.e., 1000 μg/mL), did not decrease cell viability in a statistically significant way. Moreover, it is observed that these extracts, at certain concentrations, had a positive effect on the growth of the BJ cells, enhancing their viability in comparison to the control (medium without tested extracts, 0 μg/mL). In turn, KME, KMM, and KDM extracts significantly reduced the viability of normal human skin fibroblasts only at the highest tested concentration. Interestingly, fractions obtained using ethyl acetate (KMM-O and KDM-O) were found to inhibit BJ cell viability the most potently among others, as a statistically significant decrease in cell viability was observed beginning at 250 μg/mL. Thus, this experiment showed that the toxicity of extracts derived from the *Knautia* species is highly dependent on the used extraction solution. Moreover, in this study, the cytotoxicity of caffeic acid (reference compound) was determined. The obtained results indicated that this compound possessed high ability to inhibit BJ cell viability to approx. 6% at concentration of 500 and 1000 μg/mL. Thus taking into account all results, it should be noting that KDE, KDM and KMM-B extracts seem to be the most promising and safe compounds for further biomedical applications.

In the direction of cytotoxic activity in the Caprifoliaceae family, there have been tests (e.g., with *Dipsacus asperoides*, *Cephalaria gigantea*, and *C. uralensis* species) [38,39,40].

The antiproliferative effect of a methanol extract from the roots of *Cephalaria gigantea* was tested in vitro. MEL-5 melanoma cell lines and HL-60 human leukemia cell lines were used to assess cytotoxicity. Among the tested compounds, only gigantosides D and E at a concentration of 7.5 μM had antiproliferative effects, inhibiting the growth of MEL-5 melanoma cells. Human leukemia HL-60 cells have been shown to be more sensitive to the above-mentioned saponins [39].

Cytotoxic effects were also tested for extracts from the aerial parts of *Cephalaria gigantea* and *C. uralensis* using normal skin fibroblasts (BJ cell line, ATTC CRL-2522). It was found that most of the extracts were safe for skin cells, but extracts obtained using a methanol–acetone–water mixture significantly reduced cell viability [2].

## 3. Materials and Methods

### 3.1. Chemicals and Reagents

2,2-diphenyl-1-picrylhydrazyl radical (DPPH^•^), 2,2′-azino-bis-(3-ethyl-benzothiazole-6-sulfonic acid) (ABTS^●+^), indomethacin, ascorbic acid, ethylene-diaminetetraacetic acid, disodium dihydrate (Na_2_EDTA*2H_2_O), and Trolox (±)-6-hydroxy-2,5,7,8-tetramethylchromane-2-carboxylic ac-id) were obtained from Sigma-Aldrich (Steinheim, Germany). Reference substances were supplied by ChromaDex (Irvine, CA, USA) while acetonitrile, formic acid, and water were supplied for LC analysis by Merck (Darmstadt, Germany). All other chemicals were of analytical grade and were obtained from the Polish Chemical Reagent Company (POCH, Gliwice, Poland).

### 3.2. Plant Material

The aerial parts with flowers of *Knautia drymeia* Heuff. and *Knautia macedonica* Griseb. were collected in the territory of Botanical Garden of Marie Sklodowska-Curie University in Lublin (Eastern Poland) at an altitude of 181 m a.m.s.l. coordinates N 51°15′40″ E 22°30′51″. The plants were picked up while whole plants were fully-grown in August 2021. The taxonomical identification was confirmed by Dr. Agnieszka Dąbrowska, the botanist from the Botanical Garden in Lublin.

### 3.3. Preparation of the Extracts

The plant material was dried in average temperature 24.0 ± 0.5 °C in the air and shade until constant weight was obtained [2]. Powdered plant material was extracted three times with portion of mixture of methanol–acetone–water (3:1:1; *v*/*v*/*v*) and three times with portion of 70% ethanol. With methanol–acetone–water mixture, plant material was sonicated using ultrasonic bath in controlled temperature of 40 °C (±2 °C) for 30 min every time. Extracts with 70% ethanol were macerated and shaken at room temperature for 24 h. The combined extracts were filtered and concentrated under reduced pressure. After freezing, residues were lyophilized in vacuum concentrator (Free Zone 1 apparatus; Labconco, Kansas City, KS, USA) to obtain dried crude extracts. The obtained extracts were dissolved in hot water, filtered after 24 h, and subjected to liquid–liquid extraction with ethyl acetate and n-butanol, successively. The obtained fractions were evaporated in vacuo and lyophilized using a vacuum concentrator.

### 3.4. LC-MS Analysis

The chromatographic measurements were performed using a LC-MS system from Thermo Scientific (Q-EXATCTIVE and ULTIMATE 3000, San Jose, CA, USA) equipped with an ESI source. ESI was operated in positive polarity modes under the following conditions: spray voltage—4.5 kV; sheath gas—40 arb. units; auxiliary gas—10 arb. units; sweep gas—10 arb. units; and capillary temperature—320 °C. Nitrogen (>99.98%) was employed as sheath, auxiliary, and sweep gas. The scan cycle used a full-scan event at a resolution of 70,000. A Gemini SYNERGI 4u Polar-RP column (250 × 4.6 mm, 5 μm) and a Phenomenex Security Guard ULTRA LC type guard column (the both from Phenomenex, Torrance, CA, USA) were employed for chromatographic separation, which was performed using gradient elution. The mobile phase A was 25 mM formic acid in water; the mobile phase B was 25 mM formic acid in acetonitrile. The gradient program started at 5% B increasing to 95% for 60 min; isocratic elution followed (95% B) next for 10 min. The total run time was 70 min at the mobile phase flow rate 0.5 mL/min. The column temperature was 25 °C. In the course of each run, the MS spectra in the range of 100–1000 *m*/*z* were collected continuously.

The amounts of the identified compounds were carried out based on the calibration curves obtained for the standard.

### 3.5. GC-MS Analysis

The qualification of the sample extract was performed using a GC-MS/MS system (GCMS-TQ8040; Shimadzu, Kyoto, Japan) equipped with a ZB5-MSi fused-silica capillary column (30 m × 0.25 mm i.d., 0.25 μm film thickness; Phenomenex, Torrance, CA, USA). Grade 5.0 helium was used as the carrier gas. Column flow was 1 mL/min. The injection of a 1 μL sample was performed using an AOC-20i + s type autosampler (Shimadzu, Kyoto, Japan). The injector was working at a temperature of 310 °C. The following temperature program was applied: the oven temperature was held at 60 °C for 2 min and was subsequently increased linearly at a rate of 6 °C/min to 310 °C, where it was held for 15 min. The mass spectrometer was operated in EI mode at 70 eV; the ion source temperature was 225 °C. The mass spectra were measured in the range of 40–450 amu. The amounts of the individual analytes were estimated with the peak normalization method.

### 3.6. Antioxidant Activity

All antioxidant and enzyme inhibitory assays were conducted in 96-well plates (Nunclon, Nunc, Roskilde, Denmark) using Infinite Pro 200F Elisa Reader (Tecan Group Ltd., Männedorf, Switzerland). The experiments were performed in triplicate.

#### 3.6.1. DPPH^•^ Assay

The 2,2-diphenyl-1-picryl-hydrazyl (DPPH^•^) free radical scavenging activity of *Knautia* extracts and the positive control—ascorbic acid (AA)—was examined using the method described previously [2], but with some modifications. After 30 min of incubation at 28 °C, the decrease in DPPH^•^ absorbance caused by the tested extracts was measured at 517 nm. The results were expressed as values of IC_50_.

#### 3.6.2. ABTS^●+^ Assay

The ABTS^●+^ decolorization assay was the second method applied for the assessment of antioxidant activity [2]. The absorbance was measured at 734 nm. Trolox was used as a positive control. The results were expressed as values of IC_50_.

#### 3.6.3. Metal Chelating Activity (CHEL)

The metal chelating activity was established using the method described by Guo et al. [41], modified in our previous study [2]. The absorbance was measured at 562 nm. As a positive control, Na_2_EDTA*2H_2_O was used. The results were expressed as the IC_50_ values of the *Knautia* extracts based on concentration–inhibition curves.

### 3.7. Enzyme Inhibitory Activity

#### 3.7.1. Cyclooxygenase-1 (COX-1) and Cyclooxygenase-2 (COX-2) Inhibitory Activity

The extracts of the *Knautia* species were examined for cyclooxygenase-1 (COX-1) and cyclooxygenase-2 (COX-2) inhibitory activity using a COX (ovine/human) Inhibitor Screening Assay Kit (Cayman Chemical, Ann Arbor, MI, USA) according to the protocol of the manufacturer. The extracts were tested at different concentrations. Indomethacin was used as a positive control.

#### 3.7.2. Lipoxygenase Inhibitory Activity

The anti-lipoxygenase activity of the *Knautia* extracts was determined using the Lipoxygenase Inhibitor Screening Assay Kit (Cayman Chemical, Ann Arbor, MI, USA) according to the protocol of the manufacturer. The extracts were tested at different concentrations. The effective concentration (μg/mL) in which lipoxygenase activity is inhibited by 50% (IC_50_) was estimated graphically. Indomethacin was used as a positive control.

### 3.8. Antibacterial Activity

#### 3.8.1. Bacterial Conditions

Microaerobic Gram-positive *Propionibacterium granulosum* PCM 2462 and *Cutibacterium acnes* PCM 2400 as acne strains (possessed from the Hirszfeld Institute of Immunology and Experimental Therapy, PAN, Warsaw, Poland) and aerobic Gram-positive *Staphylococcus epidermidis* ATCC 12228, *Staphylococcus aureus* ATCC 25923, and aerobic Gram-negative *Escherichia coli* ATCC 25992 bacterial strains were used. Mueller-Hinton agar or broth (MH-agar, MH-broth) for aerobic and Brain–Heart Infusion agar or broth (BHI-agar, BHI-broth) for microaerobic strains for antibacterial activity determination were applied. Inoculum was prepared by subculturing bacteria in solid medium MH or BHI at 37 °C for 24 h or 48 h, respectively. Next, inocula were prepared with fresh microbial cultures in sterile 0.9% NaCl to 0.25 McFarland turbidity standard, 0.75 × 10^8^ CFU/mL (CFU: colony forming unit).

#### 3.8.2. Disc Diffusion Method

The antibacterial activity of all extracts was performed by a modified agar disc diffusion procedure [42]. The bacterial inoculum was spread on the surface of the Petri plates containing appropriate agar, using cotton swab. The stock solutions of all tested samples (20 mg/mL) were prepared using DMSO. Next, 200 µg of extracts were placed on inoculated Petri plates. Plates with MH-agar (for aerobic strains) were incubated 24 h in 37 °C and plates with BHI-agar (for microaerobic strains) 48 h in 37 °C. The diameter of the growth inhibition zone around each sample was measured after incubation using microbiological ruler.

#### 3.8.3. MIC and MBC Determination

The minimum inhibitory concentration (MIC) of *Knautia* samples was determined for bacterial strains that exhibited the bacterial growth inhibition zones. The test was performed using the double serial microdilution in the 96-well microtiter plates according to CLSI method with some modifications [43]. Appropriate broth (200 µL) was added into each well. Double serial dilution of tested derivatives was performed in the test wells causing concentrations ranging from 6000 µg/mL to 375 µg/mL. Finally, 2 µL of tested bacteria inoculum were added to the wells (except negative sterility control). The tests were performed either at 37 °C for 24 h (aerobic strains) or 48 h (microaerobic strains). After incubation, the panel was digitally analyzed at 600 nm using the microplate reader BioTec Synergy (BioTek, Winooski, VT, USA) with a proprietary software system. The growth intensity in each well was compared with the negative and positive controls.

To determine Minimum Bactericidal Concentration (MBC), 96-well plates possessed after MIC determination were further used. Then, 10 µL of incubated medium taken from the wells in which no bacterial growth was observed were applied to the new Petri plates and incubated under optimal conditions. A visual assessment of the agar surface of the plates was made. MBC is the minimum concentration of an antimicrobial agent that is bactericidal. It is determined by subculturing medium dilutions that inhibit growth of a bacterial organism (MIC assay). The MBC was considered to be this well, in which there was no visible bacterial growth on the solid medium. To sum up the obtained results, it was benefitial to present the MBC/MIC ratio, not only MBC values. Thus, obtained MBC/MIC ratio ≤ 4 demonstrates that the drug is bactericidal. MBC/MIC values > 4 indicate the bacteriostatic nature of the tested agents [44]. Microbiological tests were performed in three separate experiments (n = 3).

### 3.9. Cytotoxic Activity

The cytotoxic activity of extracts was assessed as described in detail previously [2,45]. First, normal human skin fibroblasts (BJ cell line, ATCC CRL-2522TM) were seeded on 96-well plates (2 × 10^4^ cells/well) and maintained for 24 h at 37 °C in a humidified atmosphere. After that, two-fold serial dilutions ranged from 1000 μg/mL to 1.95 μg/mL of investigated extracts were performed. Next, BJ cells were treated with different dilutions of extracts for 72 h, and then, their viability was evaluated using MTT assay according to protocol described earlier [46].

### 3.10. Statistical Analysis

The results were expressed as mean values ± standard deviation (SD) of the indicated number of experiments. The IC_50_ values of studied extracts were calculated based on concentration–inhibition curves. Statistical significance of differences between means was established by one-way ANOVA with Dunnett’s post hoc test; *p*-values below 0.05 were considered statistically significant. The data from cell culture experiments were subjected to statistical analysis using unpaired Student’s *t*-test, and the values of the half maximum cytotoxic concentration (CC_50_) were calculated using 4-parameter nonlinear regression analyses (GraphPad Prism 5, version 5.04, Software, GraphPad Software, San Diego, CA, USA).

## 4. Conclusions

The main approach to the treatment of acne vulgaris should be complex due to the etiology of this condition. The therapeutic substance or drug should have the ability to reduce the level of sebum and inhibit the growth of bacteria. In addition, the ideal agent should also have antioxidant and anti-inflammatory properties with no toxic effect.

In our study, we accurately described the chemical composition using the LC-MS and GC-MS methods and evaluated the biological properties of the extracts and fractions prepared from the aerial parts of *Knautia drymeia* and *K. macedonica*. We have proven that both species are rich in phenolic acids, mainly cryptochlorogenic acid, chlorogenic acid, and neochlorogenic acid, as well as flavonoids, which are most likely responsible for their beneficial biological effects.

Our research provides new insight into the possibility of applying the extracts obtained from unstudied species, like *K. drymeia* and *K. macedonica*, in the treatment of inflammation-related disorders, including acne, in the adjunctive or prophylaxis therapy. The obtained results indicated that especially the ethanol and methanol–acetone–water (3:1:1, *v*/*v*/*v*) extracts from the aerial parts of *K. drymeia* showed relatively good antioxidant, anti-inflammatory, and antibacterial properties in comparison with the positive standards. Moreover, these extracts did not decrease BJ cell viability in a statistically significant way.

In conclusion, *Knautia drymeia* extracts seem to be the most promising and safe compounds for further biomedical applications; however, further in vivo clinical studies of the tested extracts are required to evaluate their modes of action and potential side effects.

## Figures and Tables

**Figure 1 ijms-24-09188-f001:**
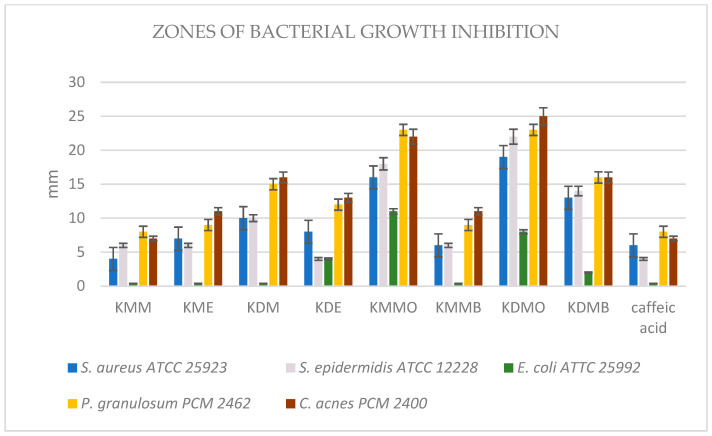
Zones of bacterial growth inhibition caused by *Knautia* extracts and fractions and caffeic acid as positive control. KDE—ethanol extract of *K. drymeia*, KDM—methanol–acetone–water (3:1:1, *v*/*v*/*v*) extract of *K. drymeia*, KME—ethanol extract of *K. macedonica*, KMM—methanol–acetone–water (3:1:1, *v*/*v*/*v*) extract of *K. macedonica,* KDM-O/KMM-O—ethyl acetate fraction, KDM-B/KMM-B—butanol fraction.

**Figure 2 ijms-24-09188-f002:**
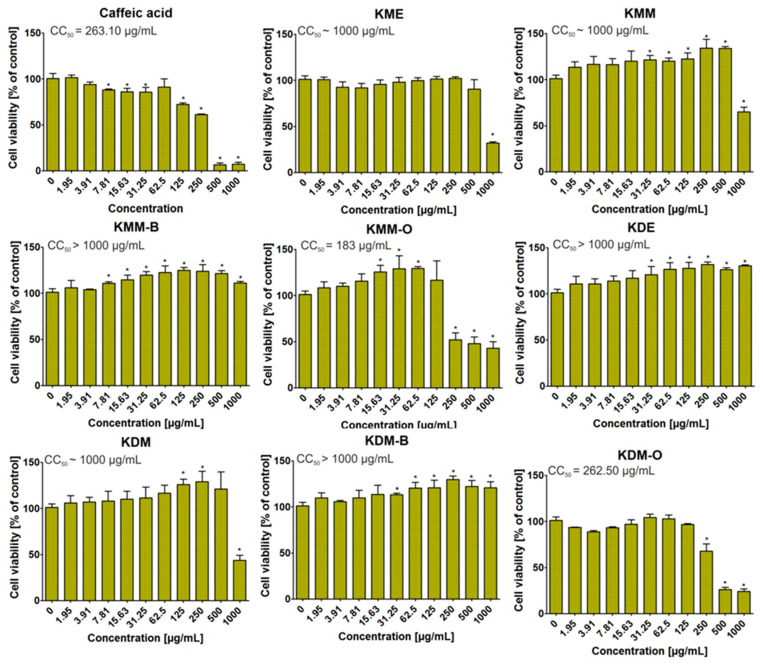
Cytotoxic activity of *Knautia* extracts and fractions towards normal human skin fibroblasts (BJ cell line, CRL-2522TM, ATCC, Manassas, VA, USA). After 72 h of incubation, cell viability was assessed using MTT assay. Caffeic acid was used as a reference compound. * Significantly different results compared to the control medium (0 μg/mL). Data analyzed using unpaired Student’s *t*-test, *p* < 0.05. KDE—ethanol extract of *K. drymeia*, KDM—methanol–acetone–water (3:1:1, *v*/*v*/*v*) extract of *K. drymeia*, KME—ethanol extract of *K. macedonica,* KMM—methanol–acetone–water (3:1:1, *v*/*v*/*v*) extract of *K. macedonica,* KDM-O/KMM-O—ethyl acetate fraction, KDM-B/KMM-B—butanol fraction.

**Table 1 ijms-24-09188-t001:** Content of active compounds in *Knautia drymeia* extracts.

Peak No.	Name of Compound	Calibration Standard	Amounts [μg/g DE]			
			KDE	KDM	KDM-O	KDM-B
1	Gallic acid	Gallic acid	68.8 ± 2.9	86.5 ± 4.1	246.0 ± 12.5	20.6 ± 0.9
2	Neochlorogenic acid	Neochlorogenic acid	5161.8 ± 247.7	3592.4 ± 176.3	5032.0 ± 245.2	3512.3 ± 161.6
3	Chlorogenic acid	Chlorogenic acid	21,845.2 ± 1133.6	24,364.3 ± 974.4	26,440.2 ± 1163.4	24,044.0 ± 1226.1
4	*p*-Hydroxybenzoic acid	*p*-Hydroxybenzoic acid	1212.5 ± 59.4	2024.6 ± 88.9	4644.9 ± 213.4	311.2 ± 13.7
5	Cryptochlorogenic acid	Cryptochlorogenic acid	21,923.6 ± 1052.2	19,562.7 ± 919.3	17,642.4 ± 846.7	18,040.1 ± 811.8
6	Caffeic acid	Caffeic acid	1884.4 ± 77.1	3948.3 ± 193.5	19,322.1 ± 792.1	391.2 ± 18.1
7	Syringic acid	Syringic acid	121.6 ± 5.2	146.3 ± 6.6	117.6 ± 4.9	12.0 ± 0.6
8	*p*-Coumaric acid	*p*-Coumaric acid	13.5 ± 0.6	30.3 ± 1.5	25.5 ± 1.2	1.9 ± 0.2
9	*o*-Coumaric acid	*p*-Coumaric acid	6.3 ± 0.3	25.3 ± 1.2	12.8 ± 0.6	2.2 ± 0.1
10	Ferulic acid	Ferulic acid	3588.8 ± 165.0	2624.7 ± 115.5	3208.8 ± 150.8	1988.7 ± 91.4
11	3,5-Di-*O*-caffeoylquinic acid	3,5-Di-*O*-caffeoylquinic acid	14,520.0 ± 653.4	9412.3 ± 423.3	21,805.9 ± 1068.2	31,560.2 ± 1325.5
12	Vanillic acid	Vanillic acid	1780.1 ± 83.7	1152.8 ± 55.3	1144.3 ± 51.5	548.0 ± 22.5
13	*m*-Hydroxybenzoic acid	*m*-Hydroxybenzoic acid	504.6 ± 22.5	363.6 ± 17.8	524.3 ± 23.1	178.1 ± 9.3
14	Salicylic acid	Salicylic acid	2220.1 ± 91.2	1260.4 ± 64.3	3112.2 ± 143.2	3072.4 ± 147.5
15	Kaempferol-3-*O*-rutinoside	Rutin	372.8 ± 17.9	892.0 ± 41.9	716.8 ± 34.4	78.8 ± 3.7
16	Rutin	Rutin	444.3 ± 18.6	507.0 ± 23.5	332.8 ± 15.6	130.7 ± 6.4
17	Luteolin-7-*O*-rutinoside	Rutin	168.4 ± 7.9	330.2 ± 16.8	374.8 ± 18.0	20.4 ± 0.8
18	Quercetin-3-*O-β*-d-(2″-*O-β*-d-xylosyl)galactoside	Rutin	23.4 ± 1.2	63.2 ± 3.0	50.8 ± 2.2	10.2 ± 0.5
19	Naringin	Rutin	54.8 ± 2.5	31.1 ± 1.4	74.4 ± 3.6	14.2 ± 0.7
20	Rhoifolin	Rutin	58.4 ± 2.7	80.8 ± 3.9	251.6 ± 12.1	62.6 ± 2.7
21	Isorhoifolin	Rutin	44.8 ± 2.2	74.4 ± 3.5	381.6 ± 19.5	50.7 ± 2.1
22	Hyperoside	Rutin	988.4 ± 46.4	1024.3 ± 43.0	2088.3 ± 108.6	1912.0 ± 78.4
23	Isoquercitrin	Rutin	148.4 ± 7.1	182.4 ± 8.2	130.4 ± 6.3	49.6 ± 2.2
24	Quercitrin	Rutin	1896.6 ± 79.6	1024.0 ± 41.2	804.2 ± 39.4	892.5 ± 44.6
25	Quercetin-3-*O*-glucuronide	Rutin	39.2 ± 1.7	395.6 ± 18.6	70.4 ± 3.3	7.6 ± 0.4
26	Orientin	Rutin	8960.0 ± 430.1	7480.0 ± 351.6	8365.4 ± 367.8	7404.4 ± 340.4
27	Homoorientin	Rutin	244.0 ± 10.2	399.2 ± 18.4	676.0 ± 30.4	208.4 ± 10.2
28	Luteolin-7-*O*-glucuronide	Rutin	305.2 ± 15.4	262.4 ± 11.5	145.6 ± 7.5	210.1 ± 9.2
29	Kaempferol-3-*O*-glucuronide	Rutin	3105.0 ± 133.3	4920.4 ± 236.2	2884.3 ± 118.2	2928.9 ± 143.5
30	Apigenin-7-*O*-glucuronide	Rutin	836.0 ± 36.8	1332.3 ± 54.6	1224.3 ± 58.6	608.8 ± 28.6
31	Luteolin-7-*O*-glucoside	Rutin	504.2 ± 22.5	1060.0 ± 51.9	1044.2 ± 45.9	153.2 ± 7.5
32	Kaempferol-7-*O*-glucoside	Rutin	99.6 ± 4.7	129.2 ± 6.2	148.8 ± 6.1	48.4 ± 2.3
33	Kaempferol-3-*O*-glucoside	Rutin	31.2 ± 1.4	60.8 ± 2.9	111.4 ± 4.8	29.3 ± 1.2
34	Isovitexin	Rutin	285.2 ± 12.0	262.4 ± 11.5	207.6 ± 9.3	261.6 ± 12.0
35	Vitexin	Rutin	1152.7 ± 55.3	2672.0 ± 136.3	2808.3 ± 123.6	1424.1 ± 69.8
36	Apigetrin	Rutin	768.8 ± 37.6	608.4 ± 28.6	327.6 ± 16.1	103.2 ± 4.9
37	Myricetin	Quercetin	12.5 ± 0.6	15.6 ± 0.6	41.6 ± 2.2	7.1 ± 0.3
38	Rosmarinic acid	Caffeic acid	772.6 ± 38.6	1912.0 ± 80.3	608.5 ± 30.4	408.0 ± 18.0
39	Sinapic acid	Ferulic acid	4.4 ± 0.2	166.0 ± 7.8	4.9 ± 0.3	0.5 ± 0.1
40	Quercetin	Quercetin	85.2 ± 4.4	80.8 ± 3.4	141.2 ± 6.4	18.2 ± 0.9
41	Kaempferol	Kaempferol	206.8 ± 9.9	448.0 ± 21.1	528.6 ± 24.8	22.1 ± 1.1
42	Luteolin	Luteolin	23.8 ± 1.1	30.2 ± 1.5	68.2 ± 2.8	6.3 ± 0.3
43	Apigenin	Apigenin	181.2 ± 8.2	341.6 ± 17.4	287.6 ± 12.4	28.7 ± 1.4
44	Naringenin	Apigenin	24.4 ± 1.0	48.4 ± 2.3	90.5 ± 4.1	24.6 ± 1.2
45	Galangin	Apigenin	17.1 ± 0.8	20.8 ± 0.9	8.6 ± 0.4	7.5 ± 0.3
46	Chrysin	Chrysin	168.4 ± 8.1	177.6 ± 7.8	127.2 ± 6.1	124.8 ± 5.5
47	Pinocembrin	Chrysin	60.8 ± 2.9	302.4 ± 13.6	79.2 ± 3.6	182.0 ± 7.3

KDE—ethanol extract of *K. drymeia*, KDM—methanol-acetone-water (3:1:1, *v*/*v*/*v*) extract of *K. drymeia*, KDM-O—ethyl acetate fraction, KDM-B—butanol fraction.

**Table 2 ijms-24-09188-t002:** Content of active compounds in *Knautia macedonica* extracts.

Peak No.	Name of Compound	Calibration Standard	Amounts [μg/g DE]			
			KME	KMM	KMM-O	KMM-B
1	Gallic acid	Gallic acid	226.2 ± 11.1	116.3 ± 5.8	476.2 ± 24.3	16.4 ± 0.7
2	Neochlorogenic acid	Neochlorogenic acid	2480.6 ± 119.0	2304.5 ± 108.3	5083.4 ± 223.5	3916.4 ± 160.6
3	Chlorogenic acid	Chlorogenic acid	21,256.1 ± 869.2	22,326.5± 1071.4	27,015.6 ± 1269.0	28,043.0 ± 1345.9
4	*p*-Hydroxybenzoic acid	*p*-Hydroxybenzoic acid	916.4 ± 44.9	1524.5 ± 64.5	6165.2 ± 301.8	315.6 ± 14.2
5	Cryptochlorogenic acid	Cryptochlorogenic acid	16,923.4 ± 710.6	26,083.5 ± 1173.6	19,280.7 ± 867.6	25,720.3 ± 1131.7
6	Caffeic acid	Caffeic acid	952.0 ± 39.0	956.4 ± 39.2	16,560.0 ± 844.6	308.4 ± 14.5
7	Syringic acid	Syringic acid	303.8 ± 13.5	364.4 ± 15.3	344.4 ± 13.8	40.4 ± 1.7
8	*p*-Coumaric acid	*p*-Coumaric acid	38.1 ± 1.6	94.4 ± 4.2	51.2 ± 2.3	4.2 ± 0.2
9	*o*-Coumaric acid	*p*-Coumaric acid	34.8 ± 1.7	100.4 ± 4.8	44.8 ± 2.1	3.8 ± 0.2
10	Ferulic acid	Ferulic acid	5960.4 ± 256.3	3264.4 ± 156.5	3776.3 ± 181.2	2812.8 ± 115.3
11	3,5-Di-*O*-caffeoylquinic acid	3,5-Di-*O*-caffeoylquinic acid	11,123.5 ± 513.4	9046.6 ± 387.1	18,885.1 ± 925.1	2188.5 ± 102.8
12	Vanillic acid	Vanillic acid	1384.0 ± 56.7	1812.4 ± 79.2	1548.6 ± 65.0	688.4 ± 28.2
13	*m*-Hydroxybenzoic acid	*m*-Hydroxybenzoic acid	148.5 ± 6.7	141.2 ± 6.4	404.6 ± 18.2	210.4 ± 9.9
14	Salicylic acid	Salicylic acid	370.4 ± 16.7	660.0 ± 31.7	1792.2 ± 75.3	484.4 ± 20.3
15	Kaempferol-3-*O*-rutinoside	Rutin	1956.0 ± 91.9	680.3 ± 30.6	796.1 ± 37.4	62.8 ± 2.8
16	Rutin	Rutin	328.8 ± 15.8	756.5 ± 33.3	1668.3 ± 80.1	816.0 ± 36.7
17	Luteolin-7-*O*-rutinoside	Rutin	712.3 ± 30.1	329.6 ± 15.5	353.6 ± 15.6	18.8 ± 0.9
18	Quercetin-3-*O-β*-D-(2″-*O-β*-D-xylosyl)galactoside	Rutin	123.2 ± 5.5	58.3 ± 2.8	60.3 ± 2.5	5.3 ± 0.2
19	Naringin	Rutin	70.4 ± 3.1	6.6 ± 0.3	25.2 ± 1.1	4.4 ± 0.2
20	Rhoifolin	Rutin	23.5 ± 1.0	58.2 ± 2.7	192.8 ± 9.3	55.2 ± 2.3
21	Isorhoifolin	Rutin	46.4 ± 2.2	34.3 ± 1.5	112.4 ± 5.3	11.2 ± 0.5
22	Hyperoside	Rutin	1516.0 ± 71.3	1368.0 ± 57.5	528.8 ± 21.6	524.0 ± 23.6
23	Isoquercitrin	Rutin	162.4 ± 7.8	215.2 ± 10.3	220.1 ± 10.6	138.8 ± 5.8
24	Quercitrin	Rutin	904.4 ± 40.7	1548.3 ± 72.8	948.2 ± 43.6	1356.0 ± 57.2
25	Quercetin-3-*O*-glucuronide	Rutin	111.2 ± 5.2	174.8 ± 8.6	1832.8 ± 80.6	127.2 ± 6.2
26	Orientin	Rutin	7423.0 ± 310.8	7680.7 ± 368.6	1832.8 ± 80.6	8084.6 ± 347.4
27	Homoorientin	Rutin	170.8 ± 7.7	852.5 ± 34.9	8564.4 ± 385.2	66.8 ± 3.2
28	Luteolin-7-*O*-glucuronide	Rutin	133.2 ± 6.3	165.2 ± 7.9	880.2 ± 42.2	322.6 ± 13.8
29	Kaempferol-3-*O*-glucuronide	Rutin	4320.4 ± 198.7	4360.4 ± 209.3	182.4 ± 8.6	5280.8 ± 216.5
30	Apigenin-7-*O*-glucuronide	Rutin	1564.0 ± 65.7	302.5 ± 12.4	4412.3 ± 198.2	2596.3 ± 124.6
31	Luteolin-7-*O*-glucoside	Rutin	992.4 ± 47.6	908.0 ± 42.7	4360.2 ± 204.9	314.4 ± 14.1
32	Kaempferol-7-*O*-glucoside	Rutin	146.0 ± 6.9	186.8 ± 9.5	2904.5 ± 133.6	65.2 ± 2.9
33	Kaempferol-3-*O*-glucoside	Rutin	23.6 ± 1.1	26.0 ± 1.2	154.5 ± 7.4	28.4 ± 1.2
34	Isovitexin	Rutin	83.2 ± 4.0	193.2 ± 9.3	108.4 ± 5.3	154.4 ± 7.3
35	Vitexin	Rutin	1512.0 ± 72.0	1668.6 ± 78.4	96.4 ± 4.9	2116.2 ± 93.1
36	Apigetrin	Rutin	668.3 ± 28.1	1532.5 ± 70.5	5840.8 ± 274.5	580.0 ± 27.8
37	Myricetin	Quercetin	13.3 ± 0.5	5.2 ± 0.2	1424.5 ± 64.1	3.2 ± 0.2
38	Rosmarinic acid	Caffeic acid	1756.5 ± 75.5	1938.0 ± 78.2	41.2 ± 2.0	1024.3 ± 52.2
39	Sinapic acid	Ferulic acid	43.6 ± 2.3	8.8 ± 0.5	1284.0 ± 59.1	1.8 ± 0.1
40	Quercetin	Quercetin	36.3 ± 1.6	78.8 ± 3.7	62.8 ± 2.8	9.5 ± 0.4
41	Kaempferol	Kaempferol	196.4 ± 9.2	263.6 ± 11.6	250.2 ± 11.8	31.7 ± 1.5
42	Luteolin	Luteolin	22.4 ± 1.0	22.3 ± 1.0	1252.5 ± 63.9	5.2 ± 0.2
43	Apigenin	Apigenin	194.4 ± 8.7	235.6 ± 11.3	78.8 ± 3.2	87.6 ± 3.7
44	Naringenin	Apigenin	60.0 ± 2.5	26.0 ± 1.1	1053.8 ± 45.3	31.3 ± 1.4
45	Galangin	Apigenin	25.7 ± 1.2	71.6 ± 2.9	86.8 ± 3.7	18.4 ± 0.9
46	Chrysin	Chrysin	198.4 ± 9.1	194.4 ± 8.2	20.6 ± 0.8	177.2 ± 8.7
47	Pinocembrin	Chrysin	912.5 ± 43.8	264.8 ± 11.4	116.8 ± 5.3	113.6 ± 5.3

KME—ethanol extract of *K. macedonica,* KMM—methanol-acetone-water (3:1:1, *v*/*v*/*v*) extract of *K. macedonica,* KMM-O—ethyl acetate fraction, KMM-B—butanol fraction.

**Table 3 ijms-24-09188-t003:** Composition of the extracts from *Knautia drymeia* (% of total fraction; mass%, GC).

Peak No.	Retention Time	Name of Compound	Area [%]			
			KDE	KDM	KDM-O	KDM-B
1	4.621	2-Ethylhexanal	nd	nd	nd	1.81
2	5.505	2-Hydroxy-γ-butyrolactone	nd	nd	nd	0.37
3	5.915	2-Ethyl-1-hexanol	nd	nd	nd	0.46
4	6.369	Glycerin	nd	nd	nd	0.72
5	6.800	2,5-Dimethylfuran-3,4(2*H*,5*H*)-dione	nd	nd	nd	0.17
6	6.948	1,4-Butanediol	nd	0.45	nd	0.23
7	7.556	Hexanoic acid, 2-ethyl-	nd	nd	nd	0.10
8	7.835	4*H*-Pyran-4-one, 2,3-dihydro-3,5-dihydroxy-6-methyl-	nd	nd	nd	0.27
9	8.390	1,2-Ethanediamine, *N*,*N*’-dibutyl-	nd	nd	nd	0.48
10	8.722	Butane, 1,1-dibutoxy-	nd	nd	nd	0.22
11	9.317	1-Butoxy-1-isobutoxy-butane	nd	nd	nd	0.11
12	9.939	Pentanoic acid, 2,2,4,4-tetramethyl-	nd	nd	nd	0.18
13	10.204	2-Methoxy-4-vinylphenol	nd	nd	3.00	nd
14	10.297	3-Penten-2-one, 3-(2-furanyl)-	nd	0.70	nd	nd
15	10.804	Eugenol	nd	0.15	nd	nd
16	11.030	1-Tridecene	0.78	nd	nd	nd
17	11.500	*cis*-7-Tetradecen-1-ol	11.06	nd	nd	nd
18	11.547	Undecane, 3-methylene-	nd	2.43	1.57	nd
19	11.552	5-Decen-1-ol, (*E*)-	nd	nd	1.96	0.87
20	11.905	Undefined compound	1.71	nd	nd	nd
21	11.988	2,4-Dodecadiene, (*E*,*Z*)-	nd	0.57	nd	0.24
22	12.007	Benzaldehyde, 2-hydroxy-6-methyl-	nd	nd	13.87	nd
23	12.170	*d*-Gluco-heptulosan	9.58	3.70	nd	nd
24	12.677	2*H*-Pyran-3(4*H*)-one, 6-ethenyldihydro-2,2,6-trimethyl-	nd	9.78	nd	nd
25	12.799	1,1-Diisobutoxy-isobutane	nd	nd	nd	2.38
26	13.365	2-Dodecenal	11.82	nd	nd	nd
27	13.477	Cyclohexane, (1,2,2-trimethylbutyl)-	nd	nd	nd	1.32
28	13.850	Undefined compound	6.08	nd	nd	nd
29	13.964	5-Dodecenol	nd	2.85	nd	nd
30	14.115	*β*-d-Glucopyranoside, methyl	34.87	56.72	31.18	nd
31	15.245	Benzeneacetic acid, 4-hydroxy-3-methoxy-, methyl ester	0.35	nd	nd	nd
32	15.355	(*E*)-4-(3-Hydroxyprop-1-en-1-yl)-2-methoxyphenol	2.06	nd	nd	nd
33	15.383	*α*-d-Glucopyranoside, methyl	nd	nd	nd	79.45
34	15.610	6-Hydroxy-4,4,7a-trimethyl-5,6,7,7a-tetrahydrobenzofuran-2(4*H*)-one	nd	nd	0.70	nd
35	15.665	1,13-Tetradecadien-3-one	5.32	nd	nd	nd
36	15.703	Tridecane, 3-methylene-	nd	2.17	nd	nd
37	16.015	Isopropyl myristate	0.53	nd	nd	nd
38	16.257	Neophytadiene	nd	0.29	1.43	nd
39	16.315	2-Pentadecanone, 6,10,14-trimethyl-	0.57	nd	nd	nd
40	16.326	2-Undecanone, 6,10-dimethyl-	nd	0.41	0.82	nd
41	17.035	Hexadecanoic acid, methyl ester	0.98	1.18	1.51	0.68
42	17.526	Palmitic acid	nd	0.58	1.03	nd
43	18.006	*β*-d-Glucosyloxyazoxymethane	nd	nd	nd	0.54
44	18.724	*trans,trans*-9,12-Octadecadienoic acid, propyl ester	nd	nd	0.19	nd
45	18.781	Linolenic acid methyl ester	nd	nd	0.47	nd
46	18.786	11(*Z*),14(*Z*),17(*Z*)-Eicosatrienoic acid methyl ester	nd	0.28	nd	0.32
47	18.894	Phytol	0.36	0.35	0.90	nd
48	18.960	Methyl stearate	1.74	1.38	2.04	1.23
49	20.140	Undec-10-ynoic acid, tetradecyl ester	0.24	0.35	nd	nd
50	20.175	E-10,13,13-Trimethyl-11-tetradecen-1-ol acetate	nd	nd	1.13	nd
51	20.406	Benzyl *β-d*-glucoside	nd	nd	nd	0.54
52	20.471	Glycidyl palmitate	nd	nd	0.17	nd
53	20.744	Methyl 18-methylnonadecanoate	nd	0.16	0.23	nd
54	22.103	Octacosanal	nd	nd	0.75	nd
55	22.155	Hexadecanoic acid, 2-hydroxy-1-(hydroxymethyl)ethyl ester	2.01	1.53	6.42	1.38
56	23.691	Methyl (*Z*)-5,11,14,17-eicosatetraenoate	nd	nd	2.16	nd
57	23.785	Octadecanoic acid, 2,3-dihydroxypropyl ester	2.20	7.11	10.18	5.10
58	25.310	Hexacosyl nonyl ether	nd	0.21	nd	nd
59	26.700	Cholesta-4,6-dien-3-ol, (3*β*)-	0.43	0.23	0.88	nd
60	27.210	2-(Decanoyloxy)propane-1,3-diyl dioctanoate	0.40	0.63	nd	nd
61	27.251	*dl-α*-Tocopherol	nd	nd	0.38	nd
62	27.268	5-Octadecenal	nd	0.20	nd	nd
63	28.013	Ergost-5-en-3-ol, (3*β*)-	nd	nd	0.40	nd
64	28.599	1-Pentacosanol	nd	nd	0.38	nd
65	28.625	Campesterol	9.16	4.70	2.76	0.18
66	28.747	Stigmast-5-ene, 3-methoxy-, (3*β*)-	nd	nd	nd	0.06
67	28.811	Cholesta-4,6-dien-3-one	nd	nd	0.88	nd
68	29.111	Acetyl betulinaldehyde	nd	nd	2.31	0.29
69	29.226	Tricyclo [5.4.3.0(1,8)]tetradecan-3-ol-9-one, 4-ethenyl-6-(2-hydroxyacetoxy)-2,4,7,14-tetramethyl-	nd	nd	1.48	nd
70	29.227	Ursolic aldehyde	nd	nd	nd	0.10
71	29.420	Stigmast-4-en-3-one	0.38	nd	nd	nd
72	29.513	Stigmasta-3,5-dien-7-one	nd	nd	0.72	nd
73	29.517	7-Oxo-5-cholesten-3*β*-yl benzoate	nd	0.34	nd	nd
74	29.610	Betulinaldehyde	nd	nd	2.79	nd
75	29.765	Cholest-4-en-26-oic acid, 3-oxo-	0.37	nd	nd	nd
76	29.851	9,19-Cyclolanostan-3-ol, 24-methylene-, (3*β*)-	nd	0.26	nd	nd
77	30.901	Methyl ursa-2,12-dien-28-oate	nd	nd	2.31	nd
78	30.903	Olean-12-en-28-oic acid, 2*β*,3*β*,23-trihydroxy-, methyl ester	nd	0.29	nd	0.20
79	31.523	Methyl 3-hydroxyurs-12-en-28-oate	nd	nd	0.93	nd
80	32.686	Uvaol	nd	nd	0.18	nd
81	33.86	Urs-12-en-28-oic acid, 2,3,19-trihydroxy-, methyl ester, (2*α*,3*β*)-	nd	nd	0.31	nd
82	33.951	Androst-5-en-7-one, 3-(acetyloxy)-, (3 *β*)-	nd	nd	0.24	nd
83	36.743	25-Nor-9,19-cyclolanostan-24-one, 3-acetoxy-24-phenyl-	nd	nd	1.34	nd

nd—not detected.

**Table 4 ijms-24-09188-t004:** Composition of the extracts from *Knautia macedonica* (% of total fraction; mass%, GC).

Peak No.	Retention Time	Name of Compound	Area [%]			
			KME	KMM	KMM-O	KMM-B
1	5.567	2-Hydroxy-γ-butyrolactone	nd	nd	nd	0.38
2	6.178	Glycerin	nd	nd	nd	1.76
3	6.824	2,5-Dimethylfuran-3,4(2*H*,5*H*)-dione	nd	nd	nd	0.1
4	6.952	1,4-Butanediol	nd	nd	nd	0.32
5	8.386	Hexanoic acid, 3-hydroxy-, methyl ester	nd	nd	nd	0.55
6	8.965	Coumaran	nd	nd	2.88	nd
7	9.101	Catechol	nd	nd	nd	1.60
8	9.194	2-Pyridinamine, 1-oxide	nd	1.32	nd	nd
9	9.361	1-Oxaspiro[3.5]nona-5,8-dien-7-one, 3-methylene-	1.73	nd	nd	nd
10	10.195	2-Methoxy-4-vinylphenol	nd	nd	0.85	1.23
11	11.329	4-Ethylcatechol	nd	nd	0.83	0.45
12	11.536	Undecanal	5.18	nd	nd	nd
13	11.548	Cyclohexene, 1,5,5-trimethyl-6-acetylmethyl-	nd	3.37	nd	nd
14	11.561	5-Decen-1-ol, (*Z*)-	nd	nd	nd	0.60
15	11.977	2,4-Dodecadiene, (*E*,*Z*)-	0.77	0.50	nd	nd
16	12.000	Benzaldehyde, 2-hydroxy-4-methyl-	nd	nd	7.62	nd
17	12.580	*d*-Gluco-heptulosan	10.24	11.49	nd	nd
18	12.804	Butane, 1,1-dibutoxy-	nd	nd	nd	9.10
19	12.804	2-Propanone, 1,1-dibutoxy-	nd	nd	nd	1.32
20	13.450	Undecane, 3-methylene-	3.79	3.12	0.54	nd
21	13.488	Tridecane, 3-methylene-	1.81	1.86	nd	0.81
22	13.896	3,7,7-Trimethylbicyclo [4.1.0]heptan-2-ol	nd	nd	1.54	nd
23	13.955	(*Z*)6-Pentadecen-1-ol	4.03	nd	nd	nd
24	13.961	Cyclopropaneacetic acid, 2-hexyl-	nd	1.17	nd	nd
25	14.202	*β*-d-Glucopyranoside, methyl	28.26	45.90	nd	64.31
26	14.359	1,2,3,5-Cyclohexanetetrol, (1α,2β,3α,5β)-	11.82	nd	21.7	nd
27	15.352	3-Hydroxymethylene-1,7,7-trimethylbicyclo [2.2.1]heptan-2-one	nd	nd	1.1	1.32
28	15.609	6-Hydroxy-4,4,7a-trimethyl-5,6,7,7a-tetrahydrobenzofuran-2(4*H*)-one	6.08	nd	0.6	nd
29	16.255	Neophytadiene	nd	0.32	1.82	nd
30	16.318	2-Undecanone, 6,10-dimethyl-	nd	nd	1.81	nd
31	16.330	6-Methyl-2-tridecanone	nd	0.74	nd	nd
32	17.131	Hexadecanoic acid, methyl ester	2.97	1.44	2.38	1.03
33	17.524	Palmitic acid	nd	0.62	1.93	nd
34	18.614	*n*-Heptadecanol-1	nd	nd	0.72	nd
35	18.718	9,12-Octadecadienoic acid (*Z*,*Z*)-, methyl ester	nd	nd	0.26	nd
36	18.729	9,12-Octadecadienoic acid, methyl ester	nd	nd	nd	0.09
37	18.730	Linoleic acid methyl ester	0.90	0.14	nd	0.27
38	18.776	11(*Z*),14(*Z*),17(*Z*)-Eicosatrienoic acid methyl ester	1.35	0.22	0.52	nd
39	18.904	Phytol	0.91	0.45	1.09	nd
40	19.011	Methyl stearate	7.80	4.73	2.77	2.98
41	19.156	2-Methyl-*Z*,*Z*-3,13-octadecadienol	nd	nd	0.35	nd
42	19.360	Octadecanoic acid	nd	nd	0.20	nd
43	20.172	Undec-10-ynoic acid, tetradecyl ester	nd	0.53	1.70	nd
44	20.735	Methyl 18-methylnonadecanoate	0.49	0.24	0.47	0.14
45	20.962	4,8,12,16-Tetramethylheptadecan-4-olide	nd	nd	0.15	nd
46	21.312	Oxalic acid, 3,5-difluorophenyl undecyl ester	nd	nd	0.35	nd
47	22.191	Pentadecanoic acid, 2-hydroxy-1-(hydroxymethyl)ethyl ester	nd	nd	5.74	nd
48	22.195	Hexadecanoic acid, 2-hydroxy-1-(hydroxymethyl)ethyl ester	2.82	2.06	9.88	1.88
49	22.496	Phthalic acid, di(6-methylhept-2-yl) ester	nd	nd	0.39	nd
50	22.508	Diisooctyl phthalate	0.33	nd	nd	nd
51	22.680	Decanoic acid, 2-hydroxy-3-[(1-oxooctyl)oxy]propyl ester	nd	0.28	nd	nd
52	23.687	2-Methyltetracosane	nd	nd	2.26	nd
53	23.711	Cinnamyl linolenate	nd	0.20	nd	nd
54	23.849	Octadecanoic acid, 2,3-dihydroxypropyl ester	10.16	9.85	1.09	7.13
55	26.042	Cholesta-4,6-dien-3-ol, (3*β*)-	nd	0.24	0.14	nd
56	26.127	Octacosane, 2-methyl-	nd	nd	0.14	nd
57	26.513	*cis*-Valerenyl acetate	nd	nd	0.26	0.07
58	26.587	Tetracyclo [5.4.3.0(7,11)]tetradeca-2,5,10-trione, 1,4,6,14-tetramethyl-4-vinyl-	nd	nd	nd	0.04
59	26.772	Stigmasta-5,22-dien-3-ol, acetate, (3*β*)-	nd	nd	1.24	nd
60	27.13	9,19-Cycloergost-24(28)-en-3-ol, 4,14-dimethyl-, acetate, (3*β*,4*α*,5*α*)-	nd	nd	0.41	nd
61	27.266	2-(Decanoyloxy)propane-1,3-diyl dioctanoate	1.45	0.72	1.36	nd
62	28.011	Lupa-13(18),22-dien-3-ol, acetate	nd	nd	0.99	nd
63	28.602	1-Heptacosanol	nd	nd	0.43	nd
64	28.739	Campesterol	12.98	6.94	8.77	0.33
65	29.111	Acetyl betulinaldehyde	0.51	0.36	2.19	0.95
66	29.226	Tricyclo [5.4.3.0(1,8)]tetradecan-3-ol-9-one, 4-ethenyl-6-(2-hydroxyacetoxy)-2,4,7,14-tetramethyl-	nd	0.22	nd	0.82
67	29.368	24-Norursa-3,12-diene	nd	0.20	0.37	nd
68	29.514	7-Oxo-5-cholesten-3*β*-yl benzoate	nd	0.49	0.74	nd
69	29.611	Khusimyl methyl ether	nd	nd	nd	1.06
70	29.619	3*β*,21*α*-diacetoxy-18,22,22-trimethyl-17,27,29,30-tetranor-c-homoolean-14-ene	nd	nd	4.76	nd
71	29.857	4-Campestene-3-one	nd	nd	0.56	nd
72	29.869	Stigmast-4-en-3-one	0.39	nd	nd	nd
73	30.898	Methyl ursa-2,12-dien-28-oate	nd	0.28	nd	0.48
74	30.899	Olean-12-en-28-oic acid, 2*β*,3*β*,23-trihydroxy-, methyl ester	1.13	nd	1.14	nd
75	31.526	Urs-12-en-28-oic acid, 3-hydroxy-, methyl ester, (3*β*)-	nd	nd	0.88	nd
76	31.529	Methyl 3-hydroxyurs-12-en-28-oate	nd	nd	nd	0.20
77	34.288	Urs-12-en-28-oic acid, 2,3,19-trihydroxy-, methyl ester, (2*α*,3*β*)-	nd	nd	0.36	nd
78	34.863	Betulinaldehyde	nd	nd	0.22	nd
79	35.743	10,12,14-Nonacosatriynoic acid	nd	nd	0.42	nd
80	36.746	Phenol, 4-[4,5-*bis*[4-(dimethylamino)phenyl]-4*H*-imidazol-2-yl]-	nd	nd	1.08	nd

nd—not detected.

**Table 5 ijms-24-09188-t005:** The IC_50_ values determined in antioxidant tests.

Sample	IC_50_		
	DPPH^●^ [mg/mL ± SD]	ABTS^●+^ [mg/mL ± SD]	CHEL [mg/mL ± SD]
KDE	1.36 ± 0.01	0.46 ± 0.07	0.15 ± 0.01
KDM	0.50 ± 0.01	0.41 ± 0.01	0.06 ± 0.01
KDM-O	2.30 ± 0.01	0.69 ± 0.02	0.26 ± 0.04
KDM-B	1.67 ± 0.02	0.50 ± 0.03	0.18 ± 0.02
KME	4.04 ± 0.02	1.25 ± 0.02	0.51 ± 0.03
KMM	3.18 ± 0.02	1.07 ± 0.05	0.40 ± 0.01
KMM-O	2.63 ± 0.02	0.85 ± 0.01	0.33 ± 0.02
KMM-B	2.20 ± 0.01	0.70 ± 0.01	0.25 ± 0.01
AA	0.48 ± 0.30	nt	nt
Trolox	nt	0.09 ± 0.10	nt
Na_2_EDTA*2H_2_O	nt	nt	0.04 ± 0.02

AA—ascorbic acid; Na_2_EDTA*2H_2_O—ethylenediaminetetraacetic acid, disodium dihydrate; nt—not tested; KDE—ethanol extract of *K. drymeia*; KDM—methanol–acetone–water (3:1:1, *v*/*v*/*v*) extract of *K. drymeia*, KME—ethanol extract of *K. macedonica,* KMM—methanol–acetone–water (3:1:1, *v*/*v*/*v*) extract of *K. macedonica*, KDM-O/KMM-O—ethyl acetate fraction, KDM-B/KMM-B—butanol fraction, ABTS—2,2′-azino-bis-(3-ethyl-benzothiazole-6-sulfonic acid); CHEL—metal chelating activity. Data were expressed as mean ± SD, n = 5.

**Table 6 ijms-24-09188-t006:** Anti-lipoxygenase and anti-cyclooxygenase activities of the extracts and fractions of *K. drymeia* and *K. macedonica*.

Sample	IC_50_ [µg/mL]		
	Lipoxygenase Inhibition	COX-1 Inhibition	COX-2 Inhibition
KDE	14.46 ± 0.92	7.63 ± 0.15	10.68 ± 0.36
KDM	27.32 ± 1.17	14.99 ± 0.21	11.65 ± 0.47
KDM-O	60.46 ± 2.01	31.97 ± 0.47	25.69 ± 0.11
KDM-B	55.86 ± 2.37	29.19 ± 0.09	24.53 ± 0.43
KME	101.19 ± 2.85	53.76 ± 1.19	76.30 ± 0.13
KMM	65.82 ± 1.79	35.02 ± 1.35	30.49 ± 0.47
KMM-O	94.23 ± 2.18	48.95 ± 1.05	44.17 ± 0.22
KMM-B	55.58 ± 1.19	30.18 ± 0.78	31.61 ± 0.35
IND	nt	4.82 ± 0.09	3.90 ± 0.02
NDGA	6.05 ± 0.07	nt	nt

KDE—ethanol extract of *K. drymeia*, KDM—methanol–acetone–water (3:1:1, *v*/*v*/*v*) extract of *K. drymeia*, KME—ethanol extract of *K. macedonica,* KMM—methanol–acetone–water (3:1:1, *v*/*v*/*v*) extract of *K. macedonica,* KDM-O/KMM-O—ethyl acetate fraction, KDM-B/KMM-B—butanol fraction, NDGA—nordihydroguaiaretic acid, IND—Indomethacin, nt—not tested. Data were expressed as mean ± SD, n = 5.

**Table 7 ijms-24-09188-t007:** Minimal inhibitory concentration (MIC) [µg/mL] of plant extracts. nt—not tested.

	*S. aureus* ATCC 25923		*S. epidermidis* ATCC 12228		*P. granulosum* PCM 2462		*C. acnes* PCM 2400	
	MIC	MBC/MIC	MIC	MBC/MIC	MIC	MBC/MIC	MIC	MBC/MIC
KMM	nt	nt	nt	nt	>6000	nt	>6000	nt
KME	nt	nt	nt	nt	>6000	nt	>6000	nt
KDM	>6000	nt	6000	>4	1500	>4	1500	>4
KDE	>6000	nt	>6000	nt	>6000	nt	6000	nt
KMM-O	3000	>4	1500	>4	1500	>4	1500	>4
KMM-B	>6000	nt	>6000	nt	6000	nt	3000	>4
KDM-O	3000	>4	1500	>4	750	>4	750	4
KDM-B	3000	>4	3000	>4	1500	>4	750	>4
Caffeic acid	nt	nt	nt	nt	>6000	nt	>6000	nt

## Data Availability

Data available on request.

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
