# Peer review of "The Anti-Acne Potential and Chemical Composition of Knautia drymeia Heuff. and Knautia macedonica Griseb Extracts"

_ijms, 2023, doi:10.3390/ijms24119188_

Round 1
Reviewer 1 Report
The presented manuscript concerns anti-acne properties of the extracts from two representatives of the genus Knautia. The experiment is well planned and described, however I have some issues to be corrected or explained:
- in the introduction some the so far known inofrmation of the tested speices should be included
- why did the authors use caffeic acid as reference compound for antibacterial assay? It seems a bit inappropriate.
- the control reference compound is missing in cytotoxicity assay
- can authors scientifically justified the use of other standard for preparing the calibration curves than the quantified compound? It is quite unusual and in my opinion it can be accepted in one or more examples, but in this work a huge number of compounds were quantified based on the calibration curve of rutin. It is not a good practice, I suppose, in terms of so many compounds.
The English is good, some minor editing needed
Reviewer 2 Report
In the present study, the authors investigated the chemical composition and biological activities significant for the anti-acne potential (antioxidant, anti-inflammatory, antimicrobial, and cytotoxic) of the extracts of two plant species, Knautia Drymeia and Knautia Macedonica. None of these species have been examined so far.
The experimental part of the work is well designed and meets the study objectives. However, the results and discussion might have been much better presented. The paper contains no discussion of the findings, merely a brief comparison with previously published findings from similar species research. I suggest writing the discussion as a separate part of the text and discussing the activities of the extracts and their fractions in relation to the chemical composition.
The list of compounds in Table 1 is repeated in Tables 2 and 3, so Table 1 is unnecessary in the text and should be transferred to the additional files, along with Figs 1, 2, 3, and 4.
Lines 365-388 should be greatly reduced because they are unimportant to the research in this paper.
Instead of 3.8.2. Zones of Bacterial Growth Inhibition should be written disc diffusion method.
Round 2
Reviewer 1 Report
The authors have improved their manuscript which now can be recommended to be published.
Reviewer 2 Report
No comments